# Incidence and patterns of adverse drug reactions among adult patients hospitalized in the University of Gondar comprehensive specialized hospital: A prospective observational follow-up study

**Ashenafi Kibret Sendekie**[1]*, **Adeladlew Kassie Netere**[1], **Samuel Tesfaye**[1], **Ephrem Mebratu Dagnew**[2], **Eyayaw Ashete Belachew**[1]

1 Department of Clinical Pharmacy, School of Pharmacy, College of Medicine and Health Sciences, University of Gondar, Gondar, Ethiopia, 2 Department of Pharmacy, College of Medicine and Health Sciences, Debre Markos University, Debre Markos, Ethiopia

* ashukib02@yahoo.com, Ashenafi.kibret@uog.edu.et

## Abstract

### Background

Adverse drug reactions (ADRs) have continued to be a public health challenge with significant clinical and healthcare costs. However, little is known regarding the incidence of ADR in Ethiopia, particularly in the study setting. Thus, this study aimed to assess the incidence and patterns of ADRs in patients admitted to the University of Gondar comprehensive specialized hospital (UoGCSH).

### Methods

A prospective observational follow-up study was conducted on admitted patients at the medical ward in the UoGCSH from May to August 2022. A multifaceted approach involving daily chart review and patient interviews was employed to collect the data. A standard Naranjo ADR Probability Scale measuring tool was used to characterize the probability of existing ADR. The data was analyzed using the Statistical Package for Social Sciences (SPSS) version 25. Logistic regression analysis was employed to determine the association between the occurrence of ADRs and other variables. A p-value at the 95% confidence interval was considered statistically significant.

### Results

This study included 237 participants in total. The average length of follow-up was 16.4 (±5.2) days. Overall, 65 ADRs were identified, resulting an incidence rate of 27.4 (95% CI: 19.8–30.4) per 100 admissions. The most common ADRs were hypokalemia (10.7%), followed by constipation, diarrhea, hypotension, and rash (9.2% each). The majority of these ADRs (73.8%) were classified as "definite" by the Naranjo ADR probability scale. Gastrointestinal

**Data Availability Statement:** All necessary files are available in the manuscript including the datasets as supporting files.

**Funding:** The author(s) received no specific funding for this work.

**Competing interests:** The authors have declared that no competing interests exist.

**Abbreviations:** ADE, Adverse drug event; ADR, Adverse drug reactions; GIT, Gastrointestinal tract; UoGCSH, University of Gondar comprehensive specialized hospital; WHO, World Health Organization.

tract (GIT) (41.5%) and metabolic (18.6%) were the most frequently exposed systems for ADR. Antibiotics (26.2%) and cardiovascular medications (24.7%) were the most frequently implicated medications in existing ADRs. ADRs were significantly associated with age (p = 0.035), the presence of comorbidities (p = 0.021) and complications (p = 0.008), and receiving a higher number of medications (p = 0.04).

## Conclusion

In this study, ADR was identified in about one-fourth of the participants. Older patients, patients with comorbidities and complications, and patients who received a higher number of medications were more likely exposed for ADRs. Healthcare providers should strictly follow the admitted patients to minimize ADRs.

## Introduction

Although clinical pharmacotherapy has a positive impact on public health, these advantages have been accompanied by an increase in the risk of medical hazards and ADRs [1, 2], which are among the most common cause of medication-related harms. The World Health Organization (WHO) defines ADR as a noxious and unintended response to medicine that occurs at doses normally used in humans for disease prevention, investigation, therapy, or for the adjustment of physiological function [3]. ADR can be either preventable or unpreventable [4]. Evidence suggests that more than half of ADR incidents can be prevented through careful medication history and follow-up which can assist the prescriber in understanding the patient's previous and current experience with the actual drug treatment [5–8].

ADR is a major concern in the healthcare system and has been a persistent issue in the health sector. Throughout medical history, ADR has affected the majority of people, caused significant morbidity and mortality, and posed a significant burden on healthcare resources [9]. There are numerous factors that predispose patients to developing ADRs, including drug-related, patient-related, infection-related, social, and adverse drug-related factors [9]. The patient's physiological and illness status influence unfavorable drug response; very young and elderly patients are more vulnerable to an unfavorable drug response than adult patients. This is usually due to a significant difference in metabolism and excretion pattern at this level and the decreased functional reserve in the extremities of age [10, 11]. Toxic effects of a compound or its metabolites may be attributed to ADR toxicity, which manifests in various organ systems, causing harmful chemical reactions, physiological dysfunction, DNA damage, or damage to cellular structures and tissues [12]. As a result, it has been suggested that ADRs be detected in various organs, such as the skin, liver, lungs, bone marrow, and kidneys [13].

ADR risk factors include age, gender, length of hospital stay, comorbidities, medication intolerance, number of medications, hereditary factor, dietary influences, environmental factors, and the skills of physicians, nurses, pharmacists, and other health professionals [5, 14–19]. Polypharmacy is unavoidable in the elderly due to multiple co-morbidities. Age-related, chronic illnesses such as dyslipidemia, hypertension, diabetes, and depression necessitate the use of multiple medications [12]. Studies have also shown that admitted patients have a higher incidence of ADR due to an increase in the number of new drug classes [20, 21]. According to a study conducted in India, 3.2% of ADRs occurred during hospitalization, and 6.2% of

hospitalization was due to ADRs [22]. Patients who had ADRs had a longer hospital stay than patients who did not have ADRs [5, 23, 24]

Health professionals have conducted numerous studies in order to mitigate the effects of ADR as much as possible; however, the efforts have not resulted in significant improvements because most ADRs are unavoidable [5]. There have been attempts by health professionals and patients to overcome these adverse drug events; however, the measures are insufficient to overcome those ADRs because more studies are needed [25]. The effects of ADR on the patient reduce the success of the patient's treatment by causing a constant failure to meet the goal of therapy and causing the patient to become entangled in another complication [21]. Despite this, the incidences and patterns of ADR are rarely studied in Ethiopia, especially in the current study setting. A study conducted in southern Ethiopia found that the incidence of ADEs was approximately 27 per 100 admissions [26], and another study found that approximately 26.6% of patients developed ADR during their hospital stay [27]. As a result, we decided to investigate the occurrences and patterns of ADRs in patients admitted to the UoGCSH in Northwest Ethiopia. Although there are several methods for detecting ADRs [28, 29], we used a patient interview, a review of medical records, and direct observation. This study will provide more information and raise awareness about the occurrence and patterns of ADRs in the study setting.

## Materials and methods

### Study design and setting

A prospective observational follow-up study was conducted among adult hospitalized patients at the medical ward of UoGCSH between May 28 and August 26, 2022. UoGCSH is one of the country's largest university hospitals, located approximately 750 kilometers from Addis Ababa, Ethiopia's capital city. The UoGCSH is a health-care facility with over 400 beds. It serves as a referral center for the area's other primary hospitals and health centers. Pediatrics, surgery, gynecology, psychiatry, HIV care, internal medical wards, and an outpatient clinic are among the many specialties available. It serves a population of five million people in the surrounding area [30].

### Study participants and inclusion criteria

All adult patients (age ≥ 18) admitted to the UoGCSH medical ward during the study period and willing to participate were included in the study. Patients who were unable to respond to the interview questions and/or had not care taker and at the same times patients with incomplete medical records which were important for assessing any change in clinical records of ADRs were excluded.

### Sample size determination and sampling technique

The sample size was calculated using the following assumptions based on the single population proportion formula: n = p (1-p) * (Z)2/d2; assuming a 5% margin error or degree of accuracy (d = 0.05), a reliability coefficient for a 95% confidence level (Z = 1.96), and a p = 0.5 (50%) response distribution for obtaining an adequate sample size. The calculated sample size was 384. Based on the hospital record, there were around a total of 655 patients admitted for the last three months. Because of this relatively small study population, we used the finite population correction formula: ni = n/(1 + n/N), n = 384/(1 + 384/655) = 242. The final sample size was 265 after accounting for the 10% expected contingency.

The study subjects were approached using a systematic random sampling technique using their unique medical record number. The first participant used as a starting point was selected using the lottery method, and then all participants were included in the study using a sampling interval. During the study period, once included, the participants were followed until they were discharged or dead.

## Data collection instruments, procedures and data quality control

The data was collected and extracted using a semi-structured data collection instrument after reviewing previous literature (S1 File). Initially, the data collection questionnaire was designed in English, translated to Amharic (local language), and then translated back to English to maintain consistency. Before actual data collection, it was pretested with 10% of randomly selected sample patients to assess the tool's quality. Finally, some changes were made before it was used to collect data. The questionnaire includes sociodemographic information, including age, gender, residence, marital status, educational level, occupation, alcohol use, and cigarette smoking, as well as clinical and related factors, including history of hospitalization in the previous six months, past medical history and current diagnosis, comorbidities, complications, and ADR history. Medications and related factors such as past and current medications, number of medications, and type of medications were also included in the questionnaire as independent variables. The Naranjo ADR probability scale was also included in the data collection instrument, and it was used to determine the likelihood that the event was caused by medication. The final questioner was the Hartwig Severity Assessment Scale questionnaire that assessed the severity of the existing ADRs.

The questionnaire was uploaded to Kobo Toolbox and used in the data collection for this study. Three clinical pharmacists volunteered to collect the data. Physicians were contacted in case of medication management changes. The investigator also followed the data collection procedures explicitly. Data was gathered through the review of patients' medical records, patient interviews, and direct observation. Prior to data collection, data collectors were trained on data collection procedures and study objectives, and they participated in participant interviews and reviewed medical charts for respective patients. They strictly monitored changes in medication experiences and abnormal laboratory values to identify the potential occurrence of ADRs. Each day of data collection, the patients' medical chart and documents such as medication orders, progress notes, laboratory results, and changes in medication experiences were assessed and strictly followed.

## ADR detection and outcome measurers

The incidence of ADR is the primary outcome of this study. The secondary outcomes are the pattern and determinants of existing ADRs. Patient interviews, patient's medical record reviews, and direct observation were used to detect the occurrence of ADR. When there is a clinically noxious and unintended response to medications for prophylaxis, investigation, or treatment, an ADR is recorded. We assessed and followed whether these symptoms of ADRs were present prior to medication initiation and whether these ADRs had well-established evidence with the suspected medications, and we approached this using both patient interviews and medical chart reviews on recorded laboratory and clinical parameters. The objective laboratory findings were also checked against the medical records of study participants suspected of having ADR. We also identified and reported the affected system and potential offending agents. The event was then classified as an ADR using expert judgment based on evidence from relevant guidelines and literature, findings from patient reports, clinical records, or laboratory results.

Most importantly, using the Naranjo ADR probability scale, we assessed and determined the extent of probability of the existing ADR being related to the suspected medications. The scale assessed the degree of the existing ADR related to the offending agent. The probability of the existing ADR was classified by expert consensus using the Naranjo ADR Probability Scale (ten questions answered as "yes," "no," or "don't know"). Each answer is assigned a different point value (-1, 0, +1, or +2). The participant's score was added up and transformed to a total score ranging from -4 to +13; the reaction was considered "definite" if it was 9 or higher, "probable" if it was 5 to 8, "possible" if it was 1 to 4, and "doubtful" if it was 0 or less [31].

Based on the Hartwig's Severity Assessment Scale, the severity of the existing ADRs were assessed and classified as mild (level 1 and 2), moderate (level 3 and 4) and severe (level 5 and above) considering factors such as requirements for change in medications, increase in hospital stay and lead to permanent injury [32].

Finally, the incidence of ADR per 100 admissions was defined as the total number of ADRs identified divided by the total number of admissions multiplied by 100. Based on the affected systems and offending agents, patterns of existing ADRs were described.

## Data quality management and statistical analysis

After the data was collected and extracted, it was entered and analyzed using the SPSS version 25. Tables and figures were used to summarize the results of descriptive statistics. The normal distribution of the data was examined using a histogram and a Q-Q plot. The mean and standard deviation were used to present continuous variables, while frequency and percent were used to present categorical variables. The association between the occurrence of ADRs and other potential independent variables was examined using a binary logistic regression analysis. Variables with a p-value of $< 0.25$ in the univariable analysis were selected for further analysis using multivariable logistic regression to identify the potential variables linked with the occurrence of ADRs. A p-value of $< 0.05$ at 95% CI was considered statistically significant.

## Research approval and participants consent

The proposal was ethically approved by the school of pharmacy ethical review committee of the University of Gondar (reference number: SOP/257/2014). Following a briefing on the study's objectives, participants were informed and written consent was obtained. The participants who were involved in the study were in a condition to provide informed consent with all proper understanding of the study purposes. All methods were carried out in accordance with relevant guidelines and regulations. The data was sufficiently anonymized and personal identifiers were not used in the data.

## Results

### Participants' sociodemographic characteristics

There were 237 participants in this study (89.4% response rate) from 265 approached patients. The majority of the participants (51.9%) were male, with a mean age of 53.0 (±17.5). The vast majority of the subjects studied (89.5%) used health insurance to cover their medical expenses (Table 1).

**Participants' clinical characteristics.** The study found that a higher proportion of the 237 participants were diagnosed with pneumonia (19%), stroke (14.3%), and heart failure (12.7%) at admission, and more than one-quarter of the patients had comorbidities (26.2%). The average duration of follow-up was 16.4 (5.2) days (Table 2).

**Table 1. Sociodemographic characteristics of hospitalized patients in the UoGCSH medical ward from May 28 to July 26, 2022 (N = 237).**

| Variables | | Frequency (%) | Mean (±SD) |
|---|---|---|---|
| Sex | Male | 123 (51.9) | |
| | Female | 114 (48.1) | |
| Age in years | 18–44 | 65 (2746) | 53.03(±17.5) |
| | 45–60 | 82 (34.6) | |
| | ≥ 61 | 90 (38) | |
| Residency | Urban | 90(38) | |
| | Rural | 147(62) | |
| Weight | | | 56.17(±9.5) |
| Educational status | Unable to read and write | 124(52.3) | |
| | Primary | 48(20.3) | |
| | Secondary | 58(24.5) | |
| | Collage and above | 7(3.0) | |
| Marital status | Single | 62(26.2) | |
| | Married | 150(63.3) | |
| | Divorced | 11(4.6) | |
| | Widowed | 14(5.9) | |
| Source of healthcare cost coverage | Health Insurance | 212(89.5) | |
| | Out of pocket | 22(9.3) | |
| | Non insurance organization | 3(1.3) | |
| Occupation | Government employee | 19(8.0) | |
| | Self-employee | 63(26.6) | |
| | Farmer | 42(17.7) | |
| | Student | 23(9.7) | |
| | Unemployed | 85(35.9) | |
| | Other | 5(2.1) | |
| Household monthly income | < 1500 | 102(43.0) | |
| | 1500–2999 | 47(19.8) | |
| | 3000–4999 | 60(25.3) | |
| | ≥ 5000 | 28(11.8) | |
| Smoking status | Currently smoking | 2(0.8) | |
| | Non-smoker at all | 224(94.5) | |
| | Previous smoker | 10(4.2) | |
| Alcohol use habits | Yes | 93(39.2) | |
| | No | 144(60.8) | |
| Physical activity | Sedentary(no) | 76(32.1) | |
| | Moderate | 135(57.0) | |
| | Vigorous | 26(11.0) | |

## Laboratory parameters of the study participants

The most deviated laboratory values observed were the levels of potassium and hemoglobin, with a mean (±SD) value of 2.8(±1.1) and 9.2 (±3.7), respectively, followed by creatinine at 1.3 (±0.56) (Table 3).

**Medications and related characteristics of the study participants.** Cardiovascular drugs (25.3%) were found to be the most commonly used class of medications prior to admission for chronic disease, with amlodipine (6.3%) overrepresenting other cardiovascular drugs, followed by antidiabetics (7.6%) and antibiotics (5.1%). Antibiotics were the most commonly prescribed

**Table 2. Clinical characteristics of hospitalized patients in the UoGCSH medical ward from May 28 to July 26, 2022 (N = 237).**

| Variables | | Frequency (%) | Mean (±SD) |
|---|---|---|---|
| Admission diagnosis | Pneumonia | 45(19) | |
| | Stroke | 34(14.3) | |
| | Heart failure | 30(12.7) | |
| | Pancytopenia | 26(11) | |
| | Tuberculosis | 25(10.5) | |
| | Meningitis | 17(7.2) | |
| | Anemia | 15(6.3) | |
| | Malaria | 14(5.9) | |
| | Diabetes | 14(5.9) | |
| | Cancer | 9(3.5) | |
| | Chronic liver disease | 9(3.5) | |
| | Chronic renal failure | 7(2.7) | |
| | Retroviral infection | 6(2.3) | |
| | Deep venous thrombosis | 4(1.4) | |
| | Poisoning | 3(1.2) | |
| | Thyroid Disorder | 3(1.2) | |
| | Others* | 12 (4.8) | |
| Number of previous admissions in the last 6 months | None | 138 (58.2) | 0.77 (±1.069) |
| | ≤ 2 | 84(35.4) | |
| | ≥ 3 | 15(6.3) | |
| Type of illness | Communicable | 58 (24.5) | |
| | Non communicable | 179 (75.5) | |
| Duration of diagnosis for chronic conditions in months | <1 | 3(6.3) | |
| | 2–11 | 20(41.6) | |
| | 12–59 | 19(39.6) | |
| | ≥60 | 6(12.5) | |
| Reason of admission for chronic patients | Exacerbation | 28(11.8) | |
| | Poor adherence | 9(3.8) | |
| | Complication | 7(3.0) | |
| | Adverse drug events | 1(0.4) | |
| Preexisting Comorbidity | Yes | 62(26.2) | |
| | No | 175(73.8) | |
| Number of comorbidities | ≤ 2 | 51(21.5) | |
| | ≥ 3 | 11(4.6) | |
| Complications of initial diagnosis | Yes | 72(30.4) | |
| | No | 165(69.6) | |
| Number of complications | ≤ 2 | 52 (21.9) | 1.3(±0.6) |
| | ≥ 3 | 20 (8.4) | |
| Average admission follow-up time in days | <7 | 44(18.6) | 16.4(±5.2) |
| | 8–14 | 54(22.8) | |
| | 15–30 | 106(44.7) | |
| | >30 | 33(13.9) | |

Others* Epilepsy, snake bite, UTI, amebiasis, BPH, severe alcohol withdrawal, bronchial asthma

**Table 3. Laboratory parameters of hospitalized patients in the UoGCSH medical ward from May 28 to July 26, 2022 (N = 237).**

| Laboratory parameters | | Frequency (%) | Mean (±SD) |
|---|---|---|---|
| CBC: | Hemoglobin | 88 (37.1) | 9.2(3.7) |
| | Hematocrit | 35 (14.8) | 32.3(10.5) |
| | Platelet | 30 (12.7) | 118(61) |
| RFT: | Serum creatinine | 67 (28.3) | 1.3(0.56) |
| | Urea | 26 (11) | 79(24.3) |
| LFT: | SGPT | 31 (13.1) | 56(6.4) |
| | SGOT | 21 (8.9) | 62(2.8) |
| Blood glucose: | Random BS | 12 (5.1) | 278 (123.7) |
| | HbA1C | 3 (1.3) | 10.3(2.9) |
| Electrolyte: | Na+ | 40 (16.9) | 132.8(0.4) |
| | K+ | 30 (12.7) | 2.8(1.1) |
| | Cl- | 12 (5.1) | 101.3(4.3) |
| Lipid profiles: | HDL | 4 (1.7) | 32(10) |
| | Total Cholesterol | 3 (1.3) | 226(47) |
| | LDL | 3 (1.3) | 128(34) |
| | Triglycerides | 1 (0.4) | 161(41) |

SGPT, serum glutamic pyruvic transferase; SGOT, glutamic-oxaloacetic transaminase; HDL, high density lipoprotein; LDL, low density lipoprotein

class of medication on admission (77.6%), followed by cardiovascular drugs (48.1%) and vitamins and minerals (10.4%) (Table 4).

## Incidence, severity and patterns of ADRs

A total of 65 adverse events were reported on 60 patients. Among a total of 237 study participants, the overall incidence of ADR was found to be 27.4 (95% CI: 19.8–30.4) per 100 admissions (Fig 1). Hypokalemia was the most common ADR (10.7%), followed by constipation, diarrhea, hypotension, and rash (9.2% each). The gastrointestinal (GIT) system was found to be the most affected (41.5%), followed by the metabolic (18.6%), cardiovascular (13.8%), and dermatological (13.8%) systems. According to the Naranjo ADR probability scale, the majority (73.8%) of existing ADRs had an average score of 9 or higher and were classified as "definite." However, the overall mean score was discovered to be 8.9(±1.0) and classified as "probable" on the ADR probability scale (Table 5 and Fig 2). Based on the severity assessment scale, majority (55.4%) of the existing ADRs were mild and 1.5% ADRs were severe (Table 6).

## Potential offending classes of medications associated with ADR

There were 65 ADRs in total. Antibiotics accounted for the most (26.2%), followed by cardiovascular (24.7%) and vitamins and minerals (13.8%). The participants experienced ADR on average within 3.5 (±3.4) days of medication initiation (Table 7).

## Adverse drug reaction management

The most common interventions used to reverse existing ADRs were medication administration (33.8%), discontinuation of the offending agents (9.2%), and dose adjustments (9.2%). Majority (89.2%) of the existed ADRs were reversed with an average of 3.6 (±3.3) days after management was initiated (Table 8).

**Table 4. Treatment pattern among hospitalized patients in the UoGCSH medical ward from May 28 to July 26, 2022 (N = 237).**

| Variables | Category | | Frequency (%) | Class frequency (%) |
|---|---|---|---|---|
| Medications were taking prior to admission | Cardiovascular drugs | Amlodipine | 15(6.3) | 60(25.3) |
| | | Enalapril | 12(4.6) | |
| | | Furosemide | 10(3.4) | |
| | | Spironolactone | 9(3) | |
| | | Hydrochlorothiazide | 8(2.5) | |
| | | Atorvastatin | 6(2.1) | |
| | Antidiabetic | Insulin | 10(3) | 18(7.6) |
| | | Metformin | 8(3) | |
| | Antibiotics | Ceftriaxone | 5(2.1) | 12(5.1) |
| | | Metronidazole | 2 (0.8) | |
| | | Cotrimoxazole | 2(0.8) | |
| | | Azithromycin | 2(0.8) | |
| | | Vancomycin | 1(0.4) | |
| | Antituberculosis | RHZE | 7(3) | 8(3.4) |
| | | Isoniazid | 1(0.4) | |
| | Anti-coagulants and anti-platelets | Aspirin | 5(2.1) | 6(2.5) |
| | | Heparin (Unfractionated) | 1(0.4) | |
| | Anti-malarial | Artesunate | 3(1.3) | 6(2.5) |
| | | Coartem | 2(0.8) | |
| | | Chloroquine | 1(0.4) | |
| Medication on admission | Antibiotics | Ceftriaxone | 69 (29.1) | 184(77.6) |
| | | Vancomycin | 34 (14.3) | |
| | | Metronidazole | 24 (10.1) | |
| | | Azithromycin | 22 (9.3) | |
| | | Ceftazidime | 9(3.8) | |
| | | Doxycycline | 9 (3.8) | |
| | | Ciprofloxacin | 6(2.5) | |
| | | Cotrimoxazole | 6(2.5) | |
| | | Ampicillin | 5(2.1) | |
| | Cardiovascular drugs | Furosemide | 39 (16.5) | 114 (48.1) |
| | | Enalapril | 20 (8.4) | |
| | | Amlodipine | 16 (6.8) | |
| | | Spironolactone | 11 (4.6) | |
| | | Hydrochlorothiazide | 9 (3.8) | |
| | | Metoprolol | 8 (3.4) | |
| | | Propranolol | 7(3) | |
| | | Atorvastatin | 4 (1.7) | |
| | Vitamins and anti-anemic agents | Iron sulphate | 24(10.1) | 80(33.8) |
| | | Cyanocobalamin | 23(9.7) | |
| | | Folic acid | 20(8.4) | |
| | | Pyridoxine | 11(4.6) | |
| | | Vit-B complex | 2 (0.4) | |
| | Anticoagulant anti platelet | Heparin (unfractionated) | 33(13.9) | 80(33.8) |
| | | Aspirin | 32(13.5) | |
| | | Clopidogrel | 8(3.4) | |
| | | Warfarin | 7(3) | |
| | Gastrointestinal agents | Omeprazole | 45(19) | 76(32.1) |
| | | Metoclopramide | 17(7.2) | |
| | | Pantoprazole | 9(3.8) | |
| | | Bisacodyl | 5(2.1) | |
| | Analgesics and antipyretic | Paracetamol | 29(12.2) | 49(20.7) |
| | | Tramadol | 15(6.3) | |
| | | Diclofenac | 3(1.3) | |
| | | Morphine | 2(0.4) | |

*(Continued)*

**Table 4.** (Continued)

| Variables | Category | | Frequency (%) | Class frequency (%) |
|---|---|---|---|---|
| Added medication during follow up | Antibiotics | Vancomycin | 15(6.3) | 55(23.2) |
| | | Ceftazidime | 13(5.5) | |
| | | Ceftriaxone | 12(5.1) | |
| | | Metronidazole | 10(4.2) | |
| | | Azithromycin | 5(2.1) | |
| | Gastrointestinal drugs | Omeprazole | 23(9.7) | 37(15.6) |
| | | Bisacodyl | 6(2.5) | |
| | | Metoclopramide | 5(2.1) | |
| | | Oral rehydration salt | 3(1.3) | |
| | Analgesics | Paracetamol | 21(8.9) | 31(13.1) |
| | | Tramadol | 10(4.2) | |
| | Vitamin and anti-anemic agents | Iron sulphate | 6(2.5) | 19(8.0) |
| | | Folic acid | 6(2.5) | |
| | | Cyanocobalamin | 5(2.1) | |
| | | Pyridoxine | 2(0.8) | |
| | Cardiovascular drugs | Enalapril | 7(3) | 16(6.8) |
| | | Amlodipine | 5(2.1) | |
| | | Metoprolol | 2(0.8) | |
| | | Atorvastatin | 2(0.8) | |
| | Anticoagulant | Heparin (unfractionated) | 9(3.8) | 12(5.1) |
| | | Warfarin | 2(0.8) | |
| | | Clopidogrel | 1(0.8) | |
| Average number of medications per patient | | $\leq 3$ | 45(19) | 5.6±1.9 |
| | | 4–5 | 101(42.6) | |
| | | $\geq 6$ | 91(38.4) | |

## Association between occurrence of ADRs and independent factors

Multivariable logistic regression analysis showed that sociodemographic and clinical characteristics of patients, such as their age, presence of comorbidities and complications, and number of medications, were found to have a significant association with the occurrence of ADRs.

Consequently, taking all other variables constant, elder patients (age $\geq 61$) were found more likely to have ADRs compared with younger patients (age 18–45) [AOR = 2.584, 95% CI: 1.081–6.046; p = 0.035]. Patients who had comorbidities [AOR = 1.945, 95% CI: 1.097–4.005; p = 0.021] and complications [AOR = 2.341, 95% CI: 1.047–7.756; p = 0.008] in addition to their initial medical conditions and who received a higher number of medications ($\geq 7$) [AOR = 2.456, 95% CI: 1.092–5.310; p = 0.04] were found more likely to be exposed for ADRs compared with those who had no comorbidities and complications and those who received lower number of medications ($\leq 3$), respectively (Table 9).

## Discussion

ADRs have a significant impact on the healthcare system's quality and the quality of life of patients. ADR monitoring could become an essential component of the healthcare system. In any case, it is frequently ignored and not considered necessary. ADR has an impact on patient treatment outcomes, which may result in multiple patients' treatment failure and medication discontinuation or abrupt adherence to medications. Failure to achieve therapy

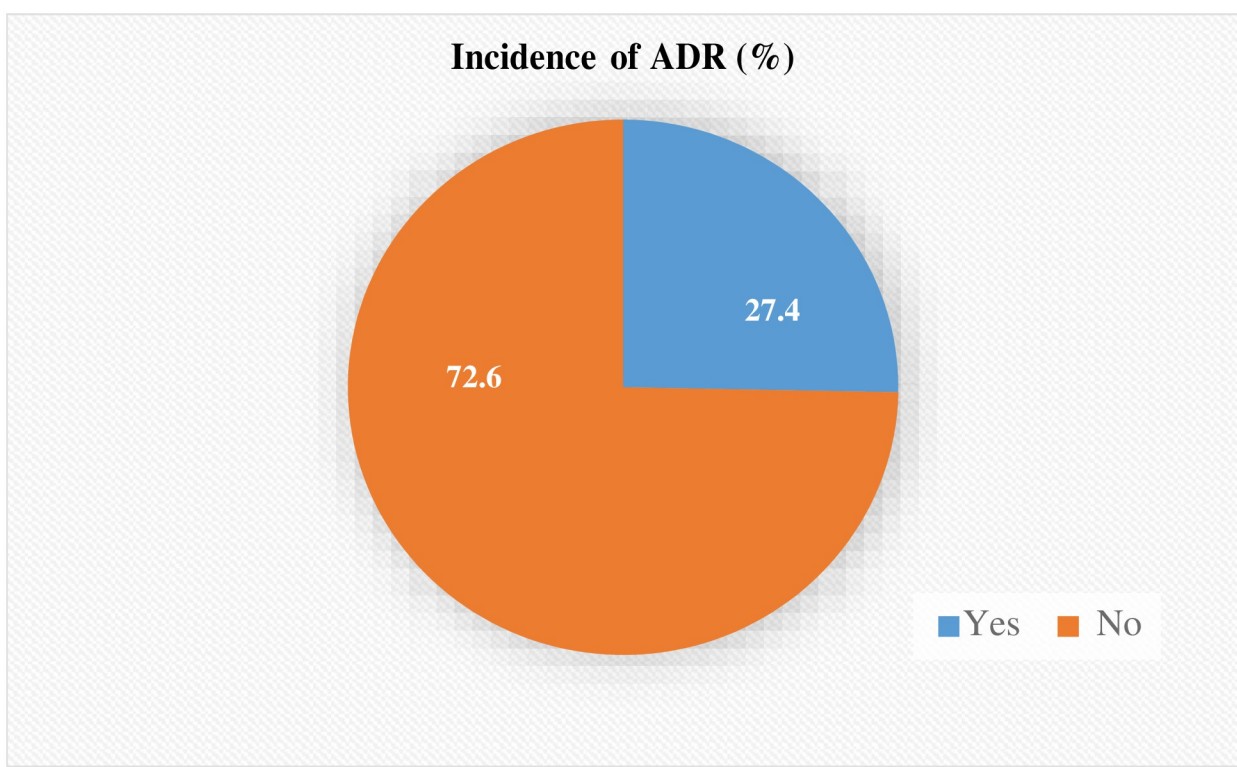

**Fig 1. Incidence of ADR among admitted patients at the UoGCSH medical ward from May 28 up to July 26, 2022 (N = 237).**

goals during the course of treatment also costs the hospital and health professionals more money to address the ADRs caused by the drugs. Health complications caused by ADRs are costly to the patient and his or her family. Furthermore, it affects the healthcare system and lowers the quality of life [33]. It should also be noted that the frequency of ADR is not

**Table 5. Patterns of adverse reactions and their Naranjo ADR probability scale among hospitalized patients in the UoGCSH medical ward from May 28 to July 26, 2022 (N = 65).**

| Systems | ADRs | Frequency (%) | Naranjo ADR mean score | Naranjo ADR Probability Scale |
|---|---|---|---|---|
| GIT | Constipation | 6 (9.2) | 10(±1.2) | Definite |
| | Diarrhea | 6(9.2) | 8(±0.9) | Probable |
| | abdominal discomfort | 5 (7.7) | 9(±1.3) | Definite |
| | Nausea and vomiting | 5 (7.7) | 10(±1.1) | Definite |
| | Dyspepsia | 5 (7.7) | 10(±1.3) | Definite |
| Metabolic | Hypokalemia | 7(10.7) | 11(±1.2) | Definite |
| | Hypoglycemia | 5(7.7) | 9(±1.0) | Definite |
| Cardiovascular | Hypotension | 6(9.2) | 9(±1.2) | Definite |
| | Edema | 3(4.6) | 8(±0.9) | Probable |
| Dermatologic | Rash | 6(9.2) | 9(±1.0) | Definite |
| | Hypersensitivity | 3(4.6) | 8(±1.1) | Probable |
| Others | Drowsiness | 3(4.6) | 9(±1.2) | Definite |
| | Bleeding | 3(4.6) | 8(±0.9) | Probable |
| | Kidney injury | 2(3.1) | 7(±0.8) | Probable |
| **Overall average Naranjo ADR score** | | | 8.9(±1.0) | Probable |

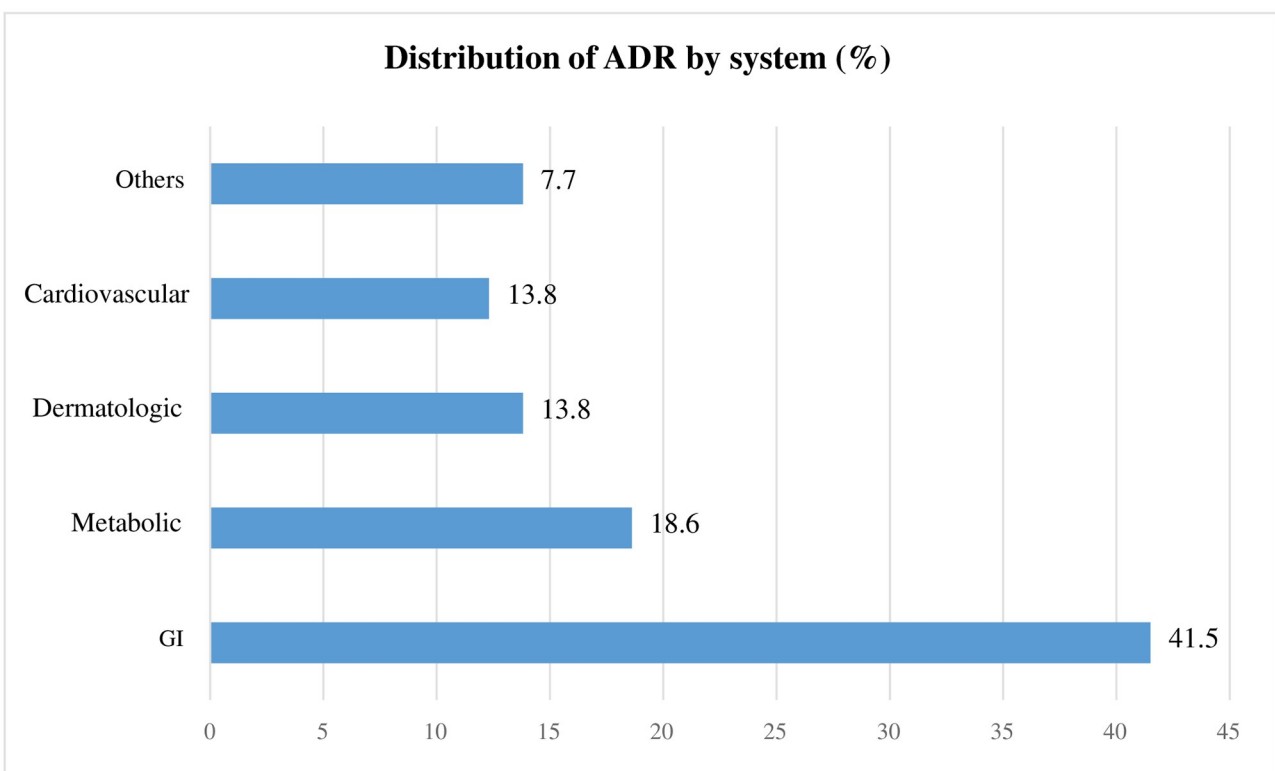

**Fig 2. Patterns of ADR respect to the affected systems among hospitalized patients in the UoGCSH medical ward from May 28 up to July 26, 2022 (N = 65).** Other, Nerve system, urinary system; GI, Gastrointestinal.

distributed evenly among all individuals. Evidence shows that it is influenced by various factors, such as age, type of medication, and duration of medication use [14, 16]. As a result, investigating these socio-demographic and clinical factors that influence the occurrence of ADR is critical. This study was carried out in this manner to assess the ADRs that occurred

**Table 6. Severity of ADRs among hospitalized patients in the UoGCSH medical ward from May 28 to July 26, 2022 (N = 65).**

| Level | Description of ADR's characteristics | Frequency (%) |
|---|---|---|
| 1 | An ADR occurred but required no change in treatment with the suspected drug | 13 (20) |
| 2 | The ADR required that treatment with the suspected drug be held, discontinued, or otherwise changed. No antidote or other treatment requirement was required. No increase in hospital stays. | 23 (35.4) |
| 3 | The ADR required that treatment with the suspected drug be held, discontinued, or otherwise changed AND/OR an antidote or another treatment was required. No increase in hospital stays. | 22 (33.9) |
| 4 | Any Level 3 ADR which increases the length of stay by at least 1 day. | 6 (9.2) |
| 5 | Any level 4 ADR which requires intensive medical care. | 1 (1.5) |
| 6 | The ADR caused permanent harm to the patient | 0 |
| 7 | The ADR which led to the death of the patient | 0 |
| Total | | 65 (100) |

Note: level 1 & 2 are mild, level 3 &4 are moderate, and level 5 and above are severe

**Table 7. Potential offending agents for the experienced ADRs (N = 65).**

| Variables | Category | Frequency (%) | Total frequency (%) |
|---|---|---|---|
| Antibiotics and anti-infective | Ceftriaxone | 7(10.8) | 17(26.2) |
| | Vancomycin | 5(7.7) | |
| | Metronidazole | 2(3.1) | |
| | Ceftazidime | 1(1.5) | |
| | Cotrimoxazole | 1(1.5) | |
| | Others | 2(3.1) | |
| Cardiovascular | Furosemide | 9(13.8) | 16(24.7) |
| | Spironolactone | 2(3.1) | |
| | Enalapril | 2(3.1) | |
| | Amlodipine | 2(3.1) | |
| | Propranolol | 1(1.5) | |
| Vitamins and anti-anemic | Cyanocobalamin | 6(9.2) | 9(13.8) |
| | Iron | 3(4.6) | |
| Analgesics | Tramadol | 6(9.2) | 8(12.3) |
| | Morphine | 2(3.1) | |
| Anticoagulants and antiplatelet | UFH | 3(4.6) | 6(9.2) |
| | Aspirin | 2(3.1) | |
| | Warfarin | 1(1.5) | |
| Anti TB | RHZE | 3(4.6) | 5(7.8) |
| | Rifampicin | 2(3.1) | |
| Others | - | 4(6.1) | 4(6.1) |
| Days since initiation of offending agent to experience ADR | $\leq 3$ | 38(58.5) | 3.5(±3.4) |
| | 4–7 | 18(27.7) | |
| | 8–14 | 6(9.2) | |
| | $\geq 15$ | 3(4.6) | |

Others, ant-leishmaniosis, anthelmintic, antimalarial

**Table 8. Adverse drug reaction management interventions (N = 65).**

| Variables | Methods of intervention | Frequency (%) | Mean(±SD) |
|---|---|---|---|
| Management interventions | Administration of medications | 22(41.5) | |
| | Dose adjustment | 6(9.2) | |
| | Discontinued | 6(9.2) | |
| | Changed | 5 (7.7) | |
| | Hold | 5(7.7) | |
| | Take with food | 2(3.1) | |
| | Slow infusion | 1(1.5) | |
| | None | 13(20) | |
| Is ADR reversed/cured | Yes | 58(89.2) | |
| | No | 7(10.8) | |
| Time taken to reverse/cure ADR | $\leq 3$ | 40(61.5) | 3.6(±3.3) |
| | 4–7 | 7(10.8) | |
| | 8–14 | 6(9.2) | |
| | $\geq 15$ | 5(7.7) | |

**Table 9. Association between occurrence of ADRs and other predicted variables.**

| Variables | | Occurrence of ADR | | 95% CI | | P-value |
|---|---|---|---|---|---|---|
| | | Yes = 65 | No = 172 | COR | AOR | |
| Sex | Male | 34 | 89 | 1.023(0.593–4.103) | 1.131(0.091–9.513) | 0.096 |
| | Female | 31 | 83 | 1 | 1 | |
| Age | ≥ 61 | 32 | 58 | 2.708(1.058–5.759) | 2.584(1.081–6.046) | 0.035* |
| | 45–60 | 22 | 60 | 1.800(0.978–8.649) | 1.345(0.876–7.645) | 0.079 |
| | 18–44 | 11 | 54 | 1 | 1 | |
| Alcohol use | Yes | 26 | 67 | 1.045(0.763–4.385) | 1.031 (0.056–7.790) | 0.137 |
| | No | 39 | 105 | 1 | 1 | |
| Physical activity | Sedentary | 21 | 55 | 1.036(0.345–5.762) | 0.785(0.023–5.231) | 0.423 |
| | Moderate | 37 | 98 | 1.025(0.471–3.084) | 1.009(0.034–6.231) | 0.321 |
| | Vigorous | 7 | 19 | 1 | 1 | |
| Type of illness | Communicable | 16 | 42 | 1.011(0.352–3.076) | 0.788(0.045–3.071) | 0.532 |
| | Non communicable | 49 | 130 | 1 | | |
| Presence of comorbidity | Yes | 22 | 40 | 1.688(1.071–3.186) | 1.945(1.097–4.005) | 0.021* |
| | No | 43 | 132 | 1 | 1 | |
| Number of medications | ≥ 7 | 29 | 62 | 2.163 (1.076–4.543) | 2.456(1.092–5.310) | 0.04* |
| | 4–6 | 28 | 73 | 1.774(0.934–3.217) | 1.672(0.813–7.334) | 0.132 |
| | ≤ 3 | 8 | 37 | 1 | 1 | |
| Presence of complications | Yes | 28 | 44 | 2.201(1.075–4.376) | 2.341(1.047–7.756) | 0.008* |
| | No | 37 | 128 | 1 | 1 | |
| Admission time in days | <7 | 12 | 32 | 1.026(0.067–5.507) | 0.932(0.014–9.713) | 0.145 |
| | 8–14 | 15 | 39 | 0.381(0.072–4.327) | 0.356(0.095–12.675) | 0.231 |
| | 15–30 | 29 | 77 | 1.004(0.043–5.538) | 0.784(0.134–10.234) | 0.452 |
| | >30 | 9 | 24 | 1 | 1 | |

AOR; Adjusted odds ratio, COR; crude odds ratio, CI; confidence interval,

* indicated p value < 0.05

in hospitalized patients at UoGCSH. Patients were interviewed, medical records were reviewed, and direct observation was used in this institutional-based observational study. The discovery yields critical information that is directly related to the patient's response to the drugs. This study's findings will also assist healthcare professionals in effectively planning to intervene with the factors related to ADRs, maximizing patients' care, and improving quality of life.

Indeed, the study participants were followed for an average of 16.4(±5.2) days. Overall, 65 adverse events were reported and resulted in an incidence rate of 27.4 (95% CI: 19.8–30.4) per 100 admissions. The most common ADRs were hypokalemia (10.7%), followed by constipation, diarrhea, hypotension, and rash (9.2% each). The majority of these ADRs (73.8%) were classified as "definite" by the Naranjo ADR probability scale and around 1.5% ADRs were severe. GIT (41.5%), metabolic (18.6%), and cardiovascular and dermatological (13.8% each) were the most frequently exposed systems for ADR. Antibiotics (26.2%), cardiovascular medications (24.7%), and vitamins and minerals (13.8%), such as iron preparations, were the most frequently implicated medications in existing ADRs. Age of the patients, presence of comorbidities and complications and number of medications were found to have a significant association with the occurrence of ADRs.

Patients, medication orders, and laboratory results were meticulously monitored throughout the study period to detect any instances of ADRs. Overall, 65 ADRs were observed in 60 patients, for an incidence rate of approximately 27.4 (95% CI: 19.8–30.4) per 100 admissions. The Naranjo ADR probability scale also revealed that majority of the existed drug ADRs were definite with an overall probable Naranjo ADR scale. This implies that ADRs are a significant burden on one-quarter of all patients admitted to adult medical wards. Several studies reported comparable incidences [26, 34, 35]. Using a similar method, Ronald K confirmed in a study of Ugandan hospitalized patients that the incidence rate of possible ADEs was 25% (95% CI: 22–29) per 100 admissions [15]. In another study, Morimoto et al from Japan found that the incidence of ADEs was 29.2% (95% CI, 27.7–30.7) per 100 admissions [36]. In consistent with an earlier study, around 1.5% of the existing ADRs were severe [26]. As the findings reveal, ADR has significant burden and requires attention, and healthcare providers must ensure patient safety.

However, a lower incidence rate was reported from Saudi Arabia, where the incidence of ADEs was reported at a rate of 8.5% (95% CI, 6.8–10.4) per 100 admissions [37]. Similarly, a lower incidence of ADR of around 8.3% was reported in other studies [38, 39]. This discrepancy might be explained by a variation in the methods employed to detect ADRs. A Saudi Arabian study relied solely on information recorded in medical records and heightened awareness by nurses. As a result, some ADE incidents that were not documented in the medical record or otherwise reported may have gone unnoticed. On the contrary, a study conducted in Uganda reported that a higher proportion of patients (48.9%) experienced at least one ADR during their hospital stay, with an incidence of 78 ADRs per 1000 person-days [40]. The discrepancy might be due to differences in methodology and study population. The current study was conducted over a short follow-up period and employed adult patients. However, the Ugandan study took longer, and it was applied to elderly patients ranging in age from 60 to 103 years.

The present study also demonstrated that, regarding systems influenced by the existing ADRs, the GIT (41.5%), metabolic system (18.6%), and cardiovascular and dermatological systems (13.8%) were the systems most affected by ADRs. The finding is in line with previous studies. According to this study, the GIT (27%), metabolic (11%), and cardiovascular (11%), were the most frequently affected systems by ADRs at the Wolaita Sodo University Teaching Referral Hospital in Ethiopia [26]. This result is consistent with another study conducted in four tertiary care public sector hospitals in Pakistan, which showed that the GIT system was frequently affected by ADR (33.3%) [41]. Similarly, a study conducted in India (51.7%) [20], Uganda (40.6%) [40], and the United States (46%) [15] revealed that GIT is a leading system affected by frequent ADRs. Another study conducted by Kumar et al reported that the GIT was found to be the most commonly affected organ system [42]. While another study showed that the metabolic and cardiovascular systems (20.8% each) are among the frequently affected systems [6]. In line with the current study, other evidence also showed that the skin is among the most affected systems by drug-induced dermatological reactions [33, 38, 43]. The findings across different studies imply that the majority of existing ADRs affect similar systems. As a result, healthcare professionals could monitor these patients prescribed with potential ADRs in the affected common systems. Potential contributing factors could be addressed, and patients need to be vigilant to avoid life-threatening ADRs.

This study also demonstrated the significance of identifying the drugs that are most likely to cause the existing ADRs. ADRs are one of the most common causes of poor treatment adherence, and assessing ADRs may help clinicians optimize drug regimens. In the current study, the prescribed medications result in varying levels of ADRs from various parts of the system. These ADRs included hypokalemia, constipation, diarrhea, hypotension, and rash, all of which are common with different classes of medications. In line with the earlier studies [14,

20, 22, 34, 38, 44], the most frequently implicated group of medicines in the ADRs were anti-microbials, followed by cardiovascular medications such as diuretics, vitamins, anti-anemics, and analgesics. Similarly, the Asosa University study found that cardiovascular and analgesic medications were among the most frequently implicated for ADRs [27]. Other evidence also revealed that cardiovascular drugs are among the most common classes of medications that result in ADRs [39, 45]. The findings revealed that the offending agents are almost identical across settings.

ADRs' causes and risk factors may be multifactorial. The current study also revealed potential risk factors for the development of ADRs. In line with the earlier studies [5, 14–19], age, the presence of comorbidities and complications, and a higher number of medications were found to have a significant association with the incidence of ADRs. This finding might suggest that older patients might be exposed to different classes of medication with different safety profiles associated with the presence of multimorbidity [12]. As a result, these patients on various medications are more likely to experience ADRs. Moreover, this study may imply that admitted patients with comorbidities and complications were also more likely to receive multiple medications with different safety profiles, which can cause ADRs. Studies have also disclosed that admitted patients with comorbidities have a higher incidence of ADR due to an increase in the number of new drug classes [20, 21]. In line with earlier study [26], the current study also reveals that patients with a higher number of medications were found to be more likely to have ADRs compared with those who received a lower number of medications. In general, older patients with a higher number of comorbidities and complications may have a higher probability of receiving multiple medications, which makes them more likely to have ADRs. As a result, admitted patients with comorbidities, complications, and a higher number of medications need to be closely monitored and followed.

To ensure medication safety, patients, caregivers, and prescribers must pay close attention to these commonly prescribed medications. Existing ADRs may also necessitate prompt intervention. The interventions for the existing ADRs ranged from managing with added medications to medication discontinuation, based on the nature of the existing ADRs. In this study, the most frequented interventions applied to reverse the existing ADRs were the administration of medications, discontinuation of the offending agents, and dose adjustments. The management interventions are consistent with previous measurements. It is also recommended to avoid the occurrence of ADRs as much as possible. Both healthcare providers and patients should be concerned about this.

In general, the current study highlighted that one-quarter of admitted patients encountered ADR. Medication errors that result in ADR can occur at any stage of medication use, including prescription dispensing, administration, and monitoring. ADRs are a significant burden that can cause serious harm, disability, and even death if they are not avoided. As a result, medications could be used, monitored, and handled correctly at every stage of the medication delivery process. The on-duty healthcare professionals pay closer attention to the clinical and laboratory manifestations of ADRs and plan patient treatment accordingly. Patients also must be extremely vigilant and motivated in order to recognize the ADR effects of their medications. Generally, the study findings are significant and can be used as a baseline for future research in the field using a population-based study.

## Study strength and limitation

The prospective follow-up of admitted patients enabled more reliable recording of medication history and symptoms, as well as assessment of causality and use of the standard scales provided. This study added additional parameters in addition to the traditional methods of

identifying ADRs based solely on self-report, such as clinical and laboratory parameters. It can contribute to a better understanding of the current incidence and pattern of ADRs in the adult medical ward at UoGCSH. In addition, the findings of this study will serve as a baseline for future researchers who wish to conduct similar research on this topic. However, it was a short-duration observational study with a small sample size. Thus, future researchers may be welcomed with a larger population size for a long period of follow-up research.

## Conclusion

This study revealed that almost one-quarter of hospitalized adults experienced ADRs during their stay in the hospital. Age, the presence of comorbidities and complications, and receiving a higher number of medications were significantly associated with the occurrence of ADRs. As a result, patients at higher risk would be followed and monitored closely. To ensure patient safety and strengthen pharmacovigilance, it is critical to raise awareness among patients, clinicians, and other staff members about reporting and preventing ADRs.

## Supporting information

**S1 File. Data collection and extraction instrument.**
(PDF)

**S2 File. Dataset: A dataset used in generating and analyzing of the data.**
(SAV)

**S1 Checklist.**
(DOC)

## Acknowledgments

The authors would like to thank the hospital administration, the word coordinators and stuffs for their positive support during the study. We would also like to forward our gratitude to the data collectors and study participants.

## Author Contributions

**Conceptualization:** Ashenafi Kibret Sendekie, Ephrem Mebratu Dagnew.

**Data curation:** Ashenafi Kibret Sendekie, Adeladlew Kassie Netere, Samuel Tesfaye, Eyayaw Ashete Belachew.

**Formal analysis:** Ashenafi Kibret Sendekie, Adeladlew Kassie Netere, Samuel Tesfaye, Ephrem Mebratu Dagnew, Eyayaw Ashete Belachew.

**Funding acquisition:** Ashenafi Kibret Sendekie.

**Investigation:** Ashenafi Kibret Sendekie, Adeladlew Kassie Netere, Ephrem Mebratu Dagnew, Eyayaw Ashete Belachew.

**Methodology:** Ashenafi Kibret Sendekie, Adeladlew Kassie Netere, Samuel Tesfaye, Ephrem Mebratu Dagnew, Eyayaw Ashete Belachew.

**Project administration:** Ashenafi Kibret Sendekie.

**Resources:** Ashenafi Kibret Sendekie, Samuel Tesfaye, Ephrem Mebratu Dagnew.

**Software:** Ashenafi Kibret Sendekie, Adeladlew Kassie Netere, Eyayaw Ashete Belachew.

**Supervision:** Ashenafi Kibret Sendekie, Eyayaw Ashete Belachew.

**Validation:** Adeladlew Kassie Netere, Eyayaw Ashete Belachew.

**Visualization:** Samuel Tesfaye, Ephrem Mebratu Dagnew.

**Writing – original draft:** Ashenafi Kibret Sendekie, Samuel Tesfaye.

**Writing – review & editing:** Ashenafi Kibret Sendekie, Adeladlew Kassie Netere, Ephrem Mebratu Dagnew, Eyayaw Ashete Belachew.

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
