## [Decision Letter · Decision Letter 0]

26 Jan 2023

PONE-D-23-00087Incidence and patterns of adverse drug reactions among adult patients hospitalized in the University of Gondar comprehensive specialized hospital: a prospective observational studyPLOS ONE

Dear Dr. Sendekie,

Thank you for submitting your manuscript to PLOS ONE. After careful consideration, we feel that it has merit but does not fully meet PLOS ONE’s publication criteria as it currently stands. Therefore, we invite you to submit a revised version of the manuscript that addresses the points raised during the review process.

We look forward to receiving your revised manuscript.

Kind regards,

Lakshmi Kannan

Academic Editor

PLOS ONE

Reviewers' comments:

Reviewer's Responses to Questions

**Comments to the Author**

1. Is the manuscript technically sound, and do the data support the conclusions?

Reviewer #1: No

Reviewer #2: Yes

Reviewer #3: Yes

Reviewer #4: Yes

2. Has the statistical analysis been performed appropriately and rigorously? 

Reviewer #1: N/A

Reviewer #2: I Don't Know

Reviewer #3: Yes

Reviewer #4: I Don't Know

3. Have the authors made all data underlying the findings in their manuscript fully available?

Reviewer #1: No

Reviewer #2: Yes

Reviewer #3: Yes

Reviewer #4: Yes

4. Is the manuscript presented in an intelligible fashion and written in standard English?

Reviewer #1: No

Reviewer #2: Yes

Reviewer #3: Yes

Reviewer #4: Yes

5. Review Comments to the Author

Reviewer #1: The major issues for this study are;

(1) Diseases and cohort vary. Thus, there is not much meaning for the conclusion.

(2) The authors did not describe disease status (i.e., those should be similar to compare).

Thus, this is a simple survey, not a scientific study.

Reviewer #2: Congratulations to all the authors for doing such a commendable job! The manuscript highlights an important topic and has been presented comprehensively in the manuscript. Wishing all the best for this amazing paper.

Reviewer #3: The authors aimed to assess the incidence and patterns of ADRs in patients admitted to the University of Gondar comprehensive specialized 6 hospitals. The authors did hard work and presented an excellent scientific report; however, this article needs these improvements to be suitable for publication.

Introduction: “The most common cause of medical harm among medication-related harms is an adverse drug reaction (ADR).” Where is the reference?

Introduction: “Evidence suggests that more than half of ADR incidents are preventable.” What do the authors mean by preventable? Do they mean that it can be prevented? If they mean that they need to write about how to prevent it?

Introduction: “This is usually due to a significant difference in metabolism and excretion pattern at this level.” Please write, “This is usually due to a significant difference in metabolism and excretion pattern at this level and the decreased functional reserve in the extremities of age.” And use this reference (https://pubmed.ncbi.nlm.nih.gov/16338704/).

Methods: ”Patients who were critically ill and unable to respond to the questions, and patients with incomplete medical records were excluded.” What about the ADR in critically ill patients, which is already high (https://pubmed.ncbi.nlm.nih.gov/21246350/) and need to be investigated?

Methods: Why do the authors not investigate the risk factors for the occurrence of these ADRs? This would be very helpful to find any correlation between the occurrence of ADRs and, for example, the socioeconomic status of the included patients.

Methods: The authors need to classify the ADRs reported as mild and moderate and series ADRs.

Results: What about the ADR reported for each class of drugs? Why did you not report which ADR had occurred with each class of drugs?

Discussion: You need to write a summary of your results at the start of the discussion.

What about ADR due to drug-drug interactions?

Provide the full form of abbreviations at the time of first use, and do not repeat writing the full form at each section.

Reviewer #4: This is an interesting study looking into the incidence of ADR in Gondar, Ethiopia.

Please clarify more on methodology of differentiating the adverse drug reaction from the effect of the disease process that led to the hospitalization.

Please clarify the language of the final questionnaire in the text as I think there are other languages spoken in Ethiopia in addition to Amharic.

More clarification on how the patient’s questionnaire was utilized to detect ADR, please clarify if it was based on signs and symptoms or on patients self-reporting of ADR. A copy from the questionnaire in English would be helpful.

Please work on making the references accurate and follow uniform citation style.

6. PLOS authors have the option to publish the peer review history of their article (what does this mean?). If published, this will include your full peer review and any attached files.

Reviewer #1: No

Reviewer #2: No

Reviewer #3: **Yes: **Omar Ahmed Abdelwahab

Reviewer #4: No

---

## [Author Response · Author response to Decision Letter 0]

30 Jan 2023

Responses to the review’s comments

Dear PLOS ONE editor,

Thank you for giving us the opportunity to submit a revised draft of the manuscript, and we would also like to thank you for your crucial comments on our paper (Manuscript ID: PONE-D-23-00087). We are very concerned and have combined all the suggested comments provided, which we believe strengthen our paper, and we hope this will render our paper eligible for consideration for publication in your reputed journal. We appreciate the time and effort that you and the reviewers dedicated to providing feedback on our manuscript and are grateful for the insightful comments and valuable improvements to our paper.

The authors would like to inform you that we have addressed the comments and recommendations made by both reviewers and the editor, point by point. In addition, throughout our revision, we made our best corrections too. All changes in the revised manuscript are highlighted using tracking changes within the manuscript. Please see below, in blue, for a point-by-point response to the reviewers’ comments and concerns. All page numbers refer to the revised manuscript file with tracked changes.

Comments from the editor:

Journal Requirements:

#1…...Please ensure that your manuscript meets PLOS ONE's style requirements, including those for file naming. 

Author response: Thank you for your recommendations to adhere PLOS ONE’s style of requirements. We have ensured that the manuscript, its name, figures, and tables are per the journal requirements.

#2…We note that you have indicated that data from this study are available upon request. PLOS only allows data to be available upon request if there are legal or ethical restrictions on sharing data publicly. 

Author response: Thank you very much for your request to revise the data availability statement. Based on your recommendation, we have revised it and stated in the main document as well as in the online submission system.

Response to Reviewers’ comments

Reviewer 1:

The major issues for this study are;

#1…...Diseases and cohort vary. Thus, there is not much meaning for the conclusion.

The authors did not describe disease status (i.e., those should be similar to compare). Thus, this is a simple survey, not a scientific study.

Author response: Thank you very much for your reviewing and comments on the manuscript, which we believe can improve the quality of this paper. 

We found both comments to be significant and similar. In fact, we shared your concern that the study participants had different medical conditions and exposures, but they were still adult patients admitted to the medical ward with comparable management care respective to their medical conditions. In short, they are an adult population admitted to medical wards. That is why we also intended to investigate the incidence and patterns of adverse drug outcomes in adults admitted to medical wards. We investigated the magnitude, types of offending agents, commonly affected systems, and extent of existing ADRs. We also believed that the findings may add some scientific evidence to existing practices for practitioners and patients, particularly in practical settings in resource-limited nations like Ethiopia. 

In addition, we have incorporated the determinants of occurrences of ADRs in revised section, which would be very helpful to identify the association between the occurrence of ADRs and possible sociodemographic and clinical characteristics of patients. 

Reviewer 2: 

Congratulations to all the authors for doing such a commendable job! The manuscript highlights an important topic and has been presented comprehensively in the manuscript. Wishing all the best for this amazing paper.

Author response: Thank you very much for your valuable review, and we also appreciate the time and effort that you dedicated to providing feedback, which we believe can improve the quality of this paper. 

Reviewer 3: 

The authors aimed to assess the incidence and patterns of ADRs in patients admitted to the University of Gondar comprehensive specialized 6 hospitals. The authors did hard work and presented an excellent scientific report; however, this article needs these improvements to be suitable for publication.

#1…. Introduction: “The most common cause of medical harm among medication-related harms is an adverse drug reaction (ADR).” Where is the reference?

Author response: Thank you very much for your comment, and we also appreciate the time and effort that you dedicated to providing feedback, which we believe can improve the quality of this paper.

Regarding the comment, we have revised and corrected the statement accordingly.

#2… Introduction: “Evidence suggests that more than half of ADR incidents are preventable.” What do the authors mean by preventable? Do they mean that it can be prevented? If they mean that they need to write about how to prevent it?

Author response: Thank you for your important suggestions and comments.

We have revised this section based on your constructive suggestions and indicated the changes with track changes in the main document.

3#...... Introduction: “This is usually due to a significant difference in metabolism and excretion pattern at this level.” Please write, “This is usually due to a significant difference in metabolism and excretion pattern at this level and the decreased functional reserve in the extremities of age.” And use this reference (https://pubmed.ncbi.nlm.nih.gov/16338704/).

Author response: We are very happy for important suggestions and corrections. After receiving your constructive comments, we have revised and incorporated the recommended reference. 

3#...... Methods: “Patients who were critically ill and unable to respond to the questions, and patients with incomplete medical records were excluded.” What about the ADR in critically ill patients, which is already high (https://pubmed.ncbi.nlm.nih.gov/21246350/) and need to be investigated?

Author response: Thank you for pointing this out. We shred your concerns, but we meant to say that participants who were unable to respond or had no caretaker for the interview questions while also having incomplete medical records that were critical for assessing any clinical records of an adverse drug reaction were excluded. Patients with varying degrees of illness severity were also included based on how they responded to interview questions or whether they had a complete medical record during follow-up periods. Therefore, we have revised it and made the context clear.

#4…. Methods: Why do the authors not investigate the risk factors for the occurrence of these ADRs? This would be very helpful to find any correlation between the occurrence of ADRs and, for example, the socioeconomic status of the included patients.

Author response: Thank you for your suggestions to include the potential contributory factors that may be linked to the occurrence of these ADRs. In this section, we have intended to determine the magnitude and patterns of ADRs. Then, in the second section, we intended to continue discussing the determinant factors. However, based on your recommendation, we found it to be much better, and we have incorporated the association between the occurrence of ADRs and other independent sociodemographic and clinical variables (Table 9).

#5...... Methods: The authors need to classify the ADRs reported as mild and moderate and series ADRs.

Author response: Thank you very much for the suggestion. As a result, we have classified and present the severity of these ADRs accordingly (Table 6). 

#6… Results: What about the ADR reported for each class of drugs? Why did you not report which ADR had occurred with each class of drugs?

Author response: Thank you for your comments and concerns. We shared your concerns and comments; the only reason we didn’t report a type of ADR for a specific class of medication is because we found that different classes of medications were involved in a single system and resulted in different ADRs. For instance, ADRs in the GIT system were related to antibiotics, vitamins, anti-anemia drugs, analgesics, and even other classes of medications. It is also true for others that the ADRs that existed in the respective systems were related to different classes of medications. Thus, we found that presenting the types of ADRs corresponding to specific classes of medication was disorganized, resulting in numerous repetitions of classes of medications and ADRs. As a result, it was found that presenting the ADR-exposed systems with their magnitudes separately and the commonly offending agents with their frequencies separately is simple and shows a concise and coherent presentation. In general, it was a matter of clear presentation; there was no other justification. If you recommend it still, we are okay with it.

#7…. Discussion: You need to write a summary of your results at the start of the discussion.

Author response: Thank you very much for your important recommendation. Thus, we have revised and presented it. 

#8…...What about ADR due to drug-drug interactions?

Author response: Thank you for your comments to clarify drug-drug interaction-related ADRs. Any identified and observed ADRs caused by any causes, including drug-drug interactions, were accounted for and included in the study's findings.

#9…...Provide the full form of abbreviations at the time of first use, and do not repeat writing the full form at each section.

Authors response: Thank you very much for your crucial comments and suggestions. Considering your valuable comments, we revised the statements accordingly.

Reviewer 4: 

This is an interesting study looking into the incidence of ADR in Gondar, Ethiopia.

#1… Please clarify more on methodology of differentiating the adverse drug reaction from the effect of the disease process that led to the hospitalization.

Author response: Thank you very much for your review and comments on the manuscript, and we also appreciate the time and effort that you dedicated to providing feedback, which we believe can improve the quality of this paper.

Regarding the comment, we shared your concern that the initial medical condition may progress in a manner similar to adverse drug reactions. However, we assessed and followed whether these symptoms of adverse drug reactions were present prior to medication initiation and whether these adverse drug reactions had well-established evidence with the suspected medications, and we approached this using both patient interviews and medical chart reviews on recorded laboratory and clinical parameters. Finally, and most importantly, using the Naranjo ADR probability scale, we assessed and determined the extent of probability of the existing adverse drug reaction being related to the suspected medications. The scale assessed the degree of the existing adverse drug reaction related to the offending agent.

As a result, we have revised it and made the context clear accordingly in the outcome measures section. 

#2...... Please clarify the language of the final questionnaire in the text as I think there are other languages spoken in Ethiopia in addition to Amharic.

 Author’s response: Thank you for your comments to clarify the language of the final questionnaire. We had used an Amharic version of the questionnaire to interview the patients because almost all of the population used Amharic in the study area. We have clarified and incorporated in the mania document accordingly. 

#3...... More clarification on how the patient’s questionnaire was utilized to detect ADR, please clarify if it was based on signs and symptoms or on patients self-reporting of ADR. A copy from the questionnaire in English would be helpful.

Author response: Thank you for your important points raised. The data collection questionnaire was designed in a semi-structured and utilized using both an interviewed based and direct observation and medical record review methods (refer page 7| ADR detection and outcome measurers). 

To detect and measure ADR, we had used both patient interviews on the existing symptoms and signs and at the same time, we also assessed and followed the medical records for potential clinical and laboratory evidences that suggest occurrence of ADRs respect to the offending agents. Based on established evidences and Naranjo probability ADR scale, we determined the extent and possibility of the ADRs related to the suspected medications. We have attached the English version data collection instrument in as supplementary material. 

4#…… Please work on making the references accurate and follow uniform citation style.

Author response: Thank you we have revised it accordingly.

---

## [Editor Report · Decision Letter 1]

7 Feb 2023

Incidence and patterns of adverse drug reactions among adult patients hospitalized in the University of Gondar comprehensive specialized hospital: a prospective observational study

PONE-D-23-00087R1

Dear Dr. Sendekie,

We’re pleased to inform you that your manuscript has been judged scientifically suitable for publication and will be formally accepted for publication once it meets all outstanding technical requirements.

Kind regards,

Lakshmi Kannan

Academic Editor

PLOS ONE

---

## [Editor Report · Acceptance letter]

15 Feb 2023

PONE-D-23-00087R1 

Incidence and patterns of adverse drug reactions among adult patients hospitalized in the University of Gondar comprehensive specialized hospital: a prospective observational follow-up study 

Dear Dr. Sendekie:

I'm pleased to inform you that your manuscript has been deemed suitable for publication in PLOS ONE. Congratulations! Your manuscript is now with our production department. 

Kind regards, 

on behalf of

Dr. Lakshmi Kannan 

Academic Editor

PLOS ONE